# Method Validation and Investigation of the Levels of Pharmaceuticals and Personal Care Products in Sludge of Wastewater Treatment Plants and Soils of Irrigated Golf Course

**DOI:** 10.3390/molecules25143114

**Published:** 2020-07-08

**Authors:** Olufemi Temitope Ademoyegun, Omobola Oluranti Okoh, Anthony Ifeanyi Okoh

**Affiliations:** 1SAMRC, Microbial Water Quality Monitoring Centre, University of Fort Hare, Alice 5700, South Africa; OOkoh@ufh.ac.za (O.O.O.); AOkoh@ufh.ac.za (A.I.O.); 2Department of Pure and Applied Chemistry, University of Fort Hare, Alice 5700, South Africa; 3National Horticultural Research Institute, P.M.B. 5432 Ibadan, Oyo State, Nigeria; 4Applied and Environmental Microbiology Research Group, Department of Biochemistry and Microbiology, University of Fort Hare, Alice 5700, South Africa

**Keywords:** method validation, optimization, sludge, soils, pharmaceutical and personal care products, irrigation golf course

## Abstract

The validation of a sensitive and reliable analytical procedure for the determination of pharmaceutical and personal care products (PPCPs) in solid environmental samples is reported in this study. Initially, two types of derivatization were used for the identification of the 13 target PPCP standards (acylation and silylation), but silylation proved to be better in sensitivity as it detected all of the analytes under investigation. Samples were extracted using an ultrasonicator, concentrated and re-dissolved in 100 mL water, then cleaned-up using C18 cartridges before silylation that preceded the Gas chromatography-mass Spectrometry detector (GC–MS) analyses. The optimized method provided a linear response over the range of 10–400 ng·g^−1^ with r^2^ > 0.992 and satisfactory recoveries (>45.6%) for the 13 compounds of interest. In this study, the variation of the sonication temperature, type of organic solvent for extraction, and types of cartridge were used to optimize the extraction procedure. A good repeatability (within day) and reproducibility (between days) with a relative standard deviation (RSD) that was equal or less than 13% for all the PPCPs were achieved with the developed extraction procedures for the irrigated soil and sewage sludge samples. The limits of detection (LODs) of the tested compounds varied from 0.1 ng·g^−1^ (aspirin) to 1.4 ng·g^−1^ (doxycycline) and from 0.1 ng·g^−1^ (codiene) to 1.7 ng·g^−1^ (doxycycline) for soils and sewage sludge samples, respectively. The method was successfully applied to the sludge of wastewater treatment plants and soils of an irrigated golf course. Among the tested emerging pollutants, paracetamol showed the highest concentration value of 98.9 ng·g^−1^ in the sludge, and for the irrigated soil (0 to 10 cm), the concentration ranged from 1.16 ng·g^−1^ (aspirin) to 8.57 ng·g^−1^ (ibuprofen).

## 1. Introduction

The incessant detection of pharmaceuticals and personal care product (PPCP) residues in the surface and drinking waters globally through the inflow of untreated and treated wastewater has brought about many enquiries, leading to the assessment of their ecological impacts in aquatic environments [1,2,3]. These groups of therapeutics called emerging contaminants are found in the receiving river waters and sediments via the discharge of effluents from wastewater treatment plants (WWTPs). Most of the plants are not efficient in the removal of these contaminants. Several investigations on pharmaceuticals that have focused on their occurrence in the influents, effluents, receiving water, drinking water, and underground sources as well as their removal efficiencies have been probed [4,5,6,7], however, studies on PPCPs in the sludge are very limited [8].

Many studies have testified that PPCPs in wastewater have been significantly reduced by sorption and easily absorb to sludge during sewage treatment processes, signifying that sludge can serve as an imperative reservoir of PPCPs [6]. Certain therapeutics, especially hydrophobic compounds with low biodegradation and limited mobility in the sludge, are predisposed to adsorb onto sewage sludge and are more unchanging than those in the wastewater [9,10]. Hence, concentration of PPCPs in sewage sludge can provide a level of indexes on their contaminants to some degree [11]. Furthermore, the disposal of sludge in landfill and their application as an agricultural manure can impact the environment with emerging contaminants [12,13]. Numerous issues have been raised about the bioactive and hydrophilic nature of emerging contaminants that pose adverse effects to the ecosystems. The effluents of WWTPs, which are one of the main pathways through which PPCPs are entering the environment, are sometimes constructed to discharge into lagoons. These are sometimes used by engineers for the construction of roads and buildings and also for the irrigation of golf fields [14]. The reuse of effluents for irrigation purposes may have adverse effects on the aquatic and terrestrial eco-systems through leaching or run-off from the soil by their exposure to low levels of the therapeutics and personal care products. Synergies of the mixture of these numerous PPCPs may exert additive effects, resulting in significant damaging impacts on wildlife and humans. Several methods are being used for the isolation of PPCPs from their various matrices and their subsequent quantification [15,16,17]. The extraction methods include sonication, Soxhlet extraction, mechanical shaking, microwave assisted extraction, and pressurized liquid extraction. There is a paucity of information in the literature on the concurrent, multi-residual identification of these groups of therapeutics using Gas chromatography with mass Spectrometry detector (GC/MS) and this area of research requires more extensive studies. Moreover, despite the fact that most of the compounds are hydrophobic, they tend to mount up on the sludge of the wastewater. There are more reported analytical methods aimed at determining these compounds in surface water and wastewater only, ignoring the possible adsorption on the sludge.

Thus, developing a method that can simultaneously determine trace level contaminants in the sewage sludge and soils and also evaluate their consequent occurrence, fate, and ecological impact is highly imperative [18]. In light of the above, the aim of this study was to optimize and validate the analytical procedures used for the simultaneous quantification of the selected PPCPs (five analgesics, three antibiotics, two anticonvulsants and personal care products, and one psychomotor stimulant in wastewater sludge and soils) in order to enhance their selectivity, sensitivity, and robustness. Therefore, the study of the occurrence and fate of these target compounds in the irrigated soils and sewage sludge in the areas cannot be over-emphasized.

## 2. Results and Discussion

### 2.1. Analytical Performance for Optimization

The dried soil and sludge used as blanks for a-SPE-GC-MS were spiked with the 10 pharmaceuticals, two personal care products, and one psychomotor stimulant. The optimization of the two solid samples was carried out with two types of cartridges (C18 and Strata X). Validation of the method was performed by spiking three different concentrations of the analyte mixture to the blank samples for the estimation of percentage recoveries. To do this, 1 mL of each level of the standard solutions containing the 13 analytes (50 ng, 100 ng, and 200 ng) in methanol was spiked to 2 g of each blank sample and the solvent (methanol) was allowed to vaporize at room temperature for about 4 min. The samples were then allowed to stand for 1 h to facilitate sorption as in nature [19]. Finally, each blank sample (sewage sludge and soil) was extracted thrice in a sonicator with 10 mL of 1:1 acetone and ethyl acetate as described in the extraction section below. Consequently, the recoveries of the analytes in the spiked sewage sludge and soil ranged from 49.6% to 89.4% and 45.6% to 106%, respectively. As a result, the two blank sample types were used to run calibration graphs for the 13 analytes. Triphenylphosphate, as the internal standard, was spiked at a 500 μg·L^−1^ concentration to ethyl acetate as the eluent. Calibration was obtained by plotting analyte-to-molecular ion peak area ratios against the analyte concentrations. The results of the offered method are shown in Table 1. As depicted, the correlation coefficients were all higher than 0.991 at seven point calibration. The limits of detection (LODs), calculated as three times the standard deviation, SD, of background noise divided by the slope of each calibration graph ranged from 0.1 to 1.4 ng·g^−1^ for soil and 0.1 to 1.7 ng·g^−1^ for sludge. While the limit of quantification (LOQs), calculated as 10 times the standard deviation of background noise divided by the slope of each calibration curve ranged from 0.3 to 4.6 ng·g^−1^ for soil and 0.3 to 5.1 ng·g^−1^ for sludge. Following successful development, the method was applied for the qualification and quantification of the target analytes in the soil samples from the golf course irrigated by Bedford WWTP effluent and sewage sludge samples taken from the other three WWTPs.

### 2.2. Quality Assurance and Quality Control

For each collection of samples, the calibration curves and two quality control (QC) samples, which were blank sludge and soils samples spiked at the limit of quantification (LOQ) (i.e., 10 times the level) were achieved. For calculation of the concentrations of the compounds in the samples, the levels measured in the extracts were adjusted by the corresponding QC recoveries. Precision of the method, expressed as relative standard deviation (RSD), was determined by five repeated injections, the with-in day (repeatability), and in between day (reproducibility). The limits of detection (LODs) and limits of quantification (LOQs) of the method for both soil and sewage sludge were calculated from spiked samples (*n* = 7), as the minimum detectable amount of the target compound with a signal-to-noise ratio of 3 and 10, respectively.

### 2.3. Optimization of Sonication-Assisted Extraction Procedure

In an attempt to optimize the potentials of sonication used for the extraction of the target PPCPs from its matrix, three parameters were put into consideration: the nature of the solvent, extraction temperature, and types of cartridges used for clean-up. The following temperatures were used for optimization: 30, 40, 50, and 60 °C, although no significant difference was observed in the recoveries of the target analytes with respect to the varied temperatures. In this study, C18 and Strata X cartridges were verified for their efficiency in one-step recovery in the cleanup of the sewage sludge extracts that were spiked at 200 ng·g^−1^ dw. As shown in Figure 1, recoveries were better using the C18 cartridge with all the therapeutics. The recoveries of all of the target compounds using C18 ranged from 70% to 106% and 31% to 88% with Strata X. Moreover, the C18 cartridge also reduced the color of the extracts, which could be linked to the background organic substances much more effectively than Strata X. Other advantages of using the C18 cartridge include a reduction of operation and sample preparation time. Therefore, the cartridge was selected for SPE clean-up of the extracts.

In this study, the recovery efficiency of the target analytes were assessed with the aforementioned combined solvents, and we observed that solvents like acetone–methanol, dichloromethane–methanol, and methanol–water combinations could also recover most of the target analytes satisfactorily. However, their recovery percentages were lower than with the acetone–ethyl acetate combination (Figure 2). Some compounds like trimethoprim and doxycycline recorded poor recoveries, as shown in Figure 2, so a combination of acetone and ethyl acetate, which yielded the optimum recovery for the selected compounds, was therefore used for other extractions in this study. The average recoveries of the analytes extracted with this solvent mixture ranged from 57.7–106.3% at the 200 ng spiked concentration that roughly concurred with the results obtained by Xu et al. (2008) [23], which ranged from 63.8 to 110.7% and had a slightly better recovery than the results by Gatidou et al. (2007) [24], which ranged from 47.6–106%. Others research studies for the optimization of solvents, probed the recovery rates for extracting target compounds from soil samples using different solvents. Their studies for natural soils and sediments spiked with PPCP mixture using microwave assisted solvent extraction with the solvent mixture of methanol:water (3:2) vol/vol indicated the optimum recovery rate [16,17]. In addition, Xu et al. (2008) [23] indicated in their study that the mixture of acetone and ethyl acetate produced better recovery for the PPCPs and some endocrine disruptive chemicals (EDCs) in sediments. Combination of acetone with methanol (1:1) vol/vol, also yielded percentage recoveries in the range of 47.6–106%, which were the best recoveries for the selected compounds [24]. Agunbiade and Moodley (2016) [25] also reported that combined acetone with 10% acetic acid-ethyl acetate solvents was the best for the extraction of target compounds in the solid sample.

### 2.4. Validation of Method

For the two types of samples (soil and sludge), precision was measured by carrying out repeatability and intermediate precision experiments. For repeatability (with-in day) experiments, six replicates of each sample were spiked at a level of 100 ng of the target compounds and analyzed during one day (*n* = 6, intra-day precision). For intermediate precision experiments, three replicates (*n* = 3) of sludge and soil samples spiked at the same concentration as above were analyzed on three different days (*k* = 3) over a period of one week (inter-day precision). Precision figures of the extraction procedure for the two types of samples are shown in Table 2. The results gave satisfactory intra- and inter-day precision of the analytical procedure for both the sludge and soil samples. RSDs were equal or less than 13% for all analytes in both substrates, signifying the good precision of the developed extraction methods for both types of samples. Azzouz and Ballesteros (2016) [26] obtained the determination of thirteen PPCPs and hormones using microwave assisted extraction (MAE), SPE, and GC/MS (RSD, 4.9–9.6%). Vazquez-roig et al. (2010) [21] developed a method for the determination of pharmaceuticals in soil and sediment using pressurized liquid extraction and LC–MS/MS, and the results showed within-day and between-day precision varying from 0.7% to 7.9% and from 1.6% to 14.5%, respectively. In our own case, a better precision was obtained after the validation exercise. To estimate the trueness of the method, recovery experiments were carried out, in which two substrate samples (about 2000 mg) were spiked at three different concentration levels for each compound as explained above (analytical performance for optimization). Table 3 shows that the recoveries ranged between 47.6% and 106% and between 49.6% and 89.4%, for the sludge and soil samples, respectively. The recoveries of the target analytes using sonication for their isolation from solid samples were comparable with the same and other extraction methods such as pressurized liquid extraction [8,27] and Soxhlet extraction Peng et al., 2006) [28] reported in the literature, but these methods are much easier and require simpler and cheaper instrumentation.

The proposed method was used to measure the 10 pharmaceuticals, two personal care products, and one psychomotor stimulant in the soil and sludge samples from different locations above-mentioned. The dried samples of the two types were analyzed in triplicate, following the analytical method mentioned under the sample extraction procedure. The box and whisker plot of the concentration ranges of the 13 PPCPs studied are presented in Figure 3. The percentage of the detection frequency of the compounds investigated of all the samples collected were as follows: ibuprofen (90), paracetamol (90), triclosan (90), caffeine (85), diclofenac (80), diethyl toluamide (75), chloramphenicol (70), codeine (70), trimethoprim (60), carbamazepine (50), diazepam (50), and doxycycline (0). The plots indicate that the highest concentrations of the analytes were observed in the sludge with ibuprofen, paracetamol, and chloramphenicol were approximately 100 ng·g^−1^ as indicated by the box and whisker plots (Figure 3). Caffeine and triclosan had their highest concentration level relatively a little above 80 ng·g^−1^ and other targeted compounds of the study showed the highest concentrations ranging from 40 to 60 ng·g^−1^. Aside from the extreme concentrations from the sludge, the data pattern of codeine, diazepam, diethyl toluamide, and aspirin showed narrow variations while the other nine PPCPs had large concentration variations. The median concentration values of the drugs showed that the relative abundance of the analytes ranked in decreasing order: carbamazepine > caffeine > triclosan > aspirin > ibuprofen > paracetamol > codeine > chloramphenicol > trimethoprim > diazepam > diethyl toluamide > diclofenac > doxycycline. Overall, the median concentration of some of the compounds were relatively lower than 20 ng·g^−1^, except for caffeine (24.32 ng·g^−1^) and codeine (20.39 ng·g^−1^). The mean concentrations of these analytes in the three WWTPs ranged from 3.40 to 98.80 ng·g^−1^. These were higher than those studied by other authors in a similar matrix, viz. 0.55–9.08 ng·g^−1^ [23], 1.43–6.57 ng·g^−1^ [21,27], and 0.52–1.99 ng·g^−1^ [29].

High level of pharmaceuticals in these types of samples were equally reported elsewhere, for instance, [8,9] determined up to 34 pharmaceutical concentrations from 1.2 to 74.9 ng·g^−1^. Dobor et al., 2010 [30] reported concentrations of 10–140 ng·g^−1^ for ibuprofen, paracetamol, aspirin, and diclofenac in sewage sludge. There was no detection of doxycycline in any of the samples analyzed in the sludge of the WWTPs, which may be due to the fact that it is not a commonly prescribed antibiotic drug in the region of study. High concentrations of PPCPs observed in the WWTPs’ sludge suggest that suspended solids have the potential ability to absorb these chemicals in influent, lessening the option of being released into surface water. Therefore, suitable disposal of sewage sludge is also important to avoid emerging contaminants re-entering the environment.

The optimized and validated procedure was also applied to soil samples from a golf course in the vicinity of the Bedford WWTP, which was irrigated twice weekly with the reclaimed wastewater. The emerging contaminants were known to be present in the effluents of the WWTPs. Four soil core samples were taken from a depth of 40 cm, each core sliced into 10 cm segments, according to the depth. Same-depth segments were combined and thoroughly mixed together. Each of the segments was determined for the target PPCPs. Figure 4 shows the concentration of the analytes in the soil profiles. In the first segment of the soil, the results showed the detection of eight selected compounds ranging from 1.16 to 8.65 ng·g^−1^ with aspirin having the lowest values and ibuprofen with the highest concentration. The second layer, from 10 to 20 cm, showed the detection of some of the compounds like ibuprofen, paracetamol, chloramphenicol, and carbamazepine, which are present in the top layer with relatively lower concentrations. The fourth layer, from 30 to 40 cm, showed no detection of the target compound and this result had the same similarity with the report of Xu et al. (2008) [23], in which the concentration levels of the PPCPs examined ranged from 0.55 to 9.08 ng·g^−1^. Lower concentration values were reported by Azzouz and Ballesteros (2012) [16] for agricultural soil, with results ranging from 9–46 ng·g^−1^. The result showed that trace emerging contaminants in the effluents of the WWTP may accumulate in the surface soils during the course of irrigation, and may eventually expose the groundwater and surface water to potential contaminants through leaching or run-off. Hence, an advanced technology for the effective removal of emerging contaminants in the WWTPs is needed before the effluents can be considered for irrigation in the environment, so as to avoid their re-introduction into the ecosystem. Therefore, pharmaceuticals are presently among the most frequently detected new classes of environmental contaminants that are present in sewage sludge and soil in contact with effluents of WWTP [31,32].

The major sources of their presence in the environment have been shown to be due to human and veterinary applications, importantly, the primary sources are the municipal WWTPs. Analgesics/anti-inflammatory drugs are among the therapeutics that are most often detected due to their frequent usage for treating common illnesses like headache and body pains [33]. Antibiotics have vital uses in both human and animal husbandry for their antibacterial possessions and as growth promoters [34]. Their presence in the soil core up to 30 cm showed that they have the ability to accumulate in the soil and sewage sludge. Anti-epileptics, which are used to relieve neuralgia, alleviate seizure disorders, and treat a wide range of mental disorders (Ramaswamy et al., 2011) [34] were also detected in the top soil. Personal care products like antiseptics are common active ingredients found in soaps, detergents, fragrances, and skin, hair, and dental care products [35]. Another one found in trace levels in the solid samples was insect repellant, which is commonly used in households for the destruction of all kinds of insects. Caffeine, found commonly in coffee and additives to food like chocolate and in pharmaceutical drugs, presented the highest concentration in the sewage sludge.

## 3. Experimental Procedures

### 3.1. Chemicals and Standards

The organic solvents used were methanol (MeOH), dichloromethane (DCM), ethyl acetate, acetone, and n-hexane, which were all High-performance liquid chromatography (HPLC) grade were purchased from (Merck, Darnstadt, Germany). The stock standards of the target compounds, internal standard (triphenylphosphate), derivatizing reagents [*N*,*O*-bis(trimethylsilyl)trifluoroacetamide (MSBSTFA)], trimethylchlorosilane (TMCS), acetic anhydride, and pyridine were from Sigma-Aldrich (Madrid, Spain); Resprep C18 and Strata X cartridges (500 mg/6 mL) (USA) were purchased from LECO. High-grade water was obtained by purifying demineralized water in a Milli-Q Gradient A10 (Millipore, Bedford, MA, USA).

Accurately weighed 0.1 g/100 mL of the individual stock solutions were prepared in methanol and kept in brown glass-stopped bottles, kept at 4 °C, and wrapped with foil paper until use. Sonication and solid phase extraction methods were used for the extraction, isolation, and pre-concentration of the compounds before the GC/MS analysis. All samples were derivatized by silylation before the chromatographic analysis. Optimization was carried out with both sludge and soils by varying the sonication temperature, extraction solvents, and cartridge types. The optimized method was thereafter applied to the sludge collected from the WWTP and soil samples from a golf course field. The information from the supervisor of the treatment plant and golf course manager suggests that for almost forty years now, the effluent has been utilized twice a week to irrigate the field. Table 4 lists the classes of therapeutic, compounds, international union of pure and allied chemistry (IUPAC) names and structures studied.

### 3.2. Sampling

Sewage sludge samples from three WWTPs using conventional activated sludge for treatment were collected monthly over a period of eight months (February to September 2017). The WWTPs were located in Alice, Seymour, and Adelaide in the Amathole District of Eastern Cape Province, South Africa. Soil samples were collected from a golf course field that had been regularly irrigated using the effluents from the Bedford WWTP for over four decades now. Sludge samples were collected in pre-cleaned amber glass (1 L) containers with lids after the final treatment of the wastewater before they were poured on the sludge bed for drying. The soil samples were collected from the golf course using a pre-cleaned hand trowel to directly scoop up into the sample jar at six different points and were pooled together. These consisted of the uppermost (approximately 10 cm deep) soils. Four core soil samples were taken from a depth of 40 cm, and each core was sliced into 10 cm segments according to the depth. All samples of the same-depth segments were pooled together and thoroughly mixed. Soil samples were obtained from 0 to 10 cm, 10 to 20 cm, 20 to 30 cm, and 30 to 40 cm depths and were stored in different glass containers. All samples were transported immediately to the laboratory in a cooler with ice blocks after collection. Soon after the sample collection, the sludge and soil samples were dried in an oven at 40 °C for two days [36], ground with a mortar and pestle, sieved with a 2-mm mesh, homogenized, and kept in a stoppered vial at −18 °C until further analyses. The soil and sludge samples that were established to contain no PPCPs in a pilot test were used as blanks, and for the optimization and validation of the method. The blank samples were subjected to the same preservation treatment as described above.

### 3.3. Extraction of Sludge and Soil Samples and Removal of Interference with Solid-Phase Extraction (SPE)

Extraction of the PPCPs from the solid samples (soils and sewage sludge) was accomplished by sonication. Initially, an appropriate amount of soil and sludge were placed in a crucible placed in an air oven at 40 °C until constant weight and was fix-dried at −18 °C. Two grams of the dried blank sample was spiked with the methanolic mixed solution of the standards of the thirteen compounds (50 ng each) and left in a fume cupboard for 1 h to remove the organic solvent. The sonication was carried out thrice at 30 °C for 30 min using a 10 mL mixture of acetone and ethyl acetate (1:1 *v*/*v*) as the extraction solvent. The supernatants after three extractions were pooled together, centrifuged, concentrated with a rotary evaporator, and later to 1 mL by nitrogen gas and reconstituted to 100 mL of distilled water of Milli-Q grade [37]. One milliliter of the freshly prepared internal standard in ethyl acetate (500 µg·L^−1^) was added to the solution, after which it was passed through cartridges that were conditioned successively by 5 mL of methanol and 5 mL of Milli-Q grade water at a flow rate of 0.5 mL·min^−1^. The samples were allowed to flow in to the cartridge at a rate of 10 mL·min^−1^. In order to remove any interference, 10 mL of Milli-Q grade water was used to wash the cartridges and then dried under vacuum for 30 min. Furthermore, 5 mL of ethyl acetate was used to elute the compounds and the eluates were evaporated to dryness under a gentle stream of nitrogen [16]. Finally, the dried residue was reconstituted with 100 µL of n-hexane prior to the derivatization reaction.

### 3.4. Derivatization Procedure

PPCPs are often derivatized before GC/MS analysis because of their non-detection and/or low volatility [38]. This procedure is important in order to realize the highest sensitivity of the target compounds. There are quite a few derivatization methods such as acylation, alkylation, and silylation that have been used to enhance the detection of pharmaceutical active compounds in GC/MS. Preliminary experiments were carried out in order to choose between two methods: acylation and silylation. Table 5 shows the results of the derivatization of the compounds, the analyte detection, retention times, and the chemical abstracts service (CAS) number for each compound.

For silylation, a volume of 200 µL of BSTFA containing 1% TMCS was added to the vial containing 100 µL of the reconstituted residue. The vial was closed and totally mixed for 1 min using a vortex machine. The derivatization reaction was achieved at 70 °C for 20 min [26]. The derivatized extract was permitted to cool to room temperature before injection into the GC-MS for identification and quantification. The acylation was performed the same way using acetic acid plus 1% pyridine. The results showed that more target compounds were detected by silylation with narrower and higher chromatographic peaks and better percentage relative standard deviation (% RSD) of the retention time than acylation (Table 5). As reported, silylation reacts very quickly with hydroxyl group compounds, succeeding derivatized compounds with volatility, adequate stability, and solubility and it is volatile enough to elute very quickly near the void volume, and therefore, it is the most widespread analytical procedure for enhancing the chromatographic separation of low-volatile compounds by gas chromatography [39,40,41,42]. In previous studies, Azzouz et al., (2010) [37] demonstrated the use of BSTFA plus 1% TMCS as a catalyst for complete derivatization, and this was employed in this study.

In auto-sampling, the issue of the sample volume must be considered. In a 1000 µL amber bottle used for auto sampling by the GC/MS, it was observed that any volume below 250 µL could not be reached by the auto-injection needle. Therefore, the final volume was increased by increasing the volume of the solvent used for extract reconstitution and derivatization reagent (BSTFA) in the ratio mentioned below to realize the complete derivatization of the target analytes. One hundred microliters (100 µL) of each of the following solvents: acetone, n-hexane, acetonitrile, methanol, and ethyl acetate, with only BSTFA plus 1% TMCS were used in duplicate in each case to check for complete derivatization. Of the five solvents, reconstitution with n-hexane and acetonitrile yielded complete derivatization. This allowed us to increase the sample volume to about 300 µL (100 µL of hexane plus 200 µL of derivatization reagent) to avoid void injection of samples into the GC/MS.

### 3.5. Gas Chromatography-Mass Spectrophotometry Analysis

For the detection of PPCPs, an Agilent 5977A (single quadrupole) mass spectrometer connected with a gas chromatograph (Agilent 7980B) was used. The GC was equipped with a HP-5 fused silica capillary column (30 m × 0.25 mm × 0.25 µm) coated with 5% phenylmethyl-polysiloxane. Helium, which moved at flow rate of 1 mL per min, was the carrier gas. After preliminary experiments, the column temperature was programmed as follows: the oven temperature was set at 70 °C for 1 min following injection and ramped to 150 °C at 14 °C·min^−1^; then from 150 to 290 °C at 6 °C·min^−1^. The injection port and transfer line temperatures were maintained at 270 and 280 °C, respectively, whereas the ion source temperature was 180 °C. For identification of the target compounds, the full scan mode was used, setting the mass range from 50 to 400. Additionally, the relative retention time of each compound was used as an additional tool for the confirmation of the presence of the compounds in the unknown samples. Table 1 shows the molecular weight, compound names, retention times, CAS numbers, and the relative standard deviations of the target compounds. The selected ions of the compounds after acetylation and trimethylsilylation are in agreement with those reported elsewhere [16,23,38,40].

## 4. Conclusions

The validated method gave acceptable recovery efficiencies for all of the target analytes in the two different substrate matrices. It provided an analytical tool to study the presence of selected compounds in any solid matrix in the environment. The presence of PPCPs in the two substrates were simultaneously determined by ultrasonic assisted solvent extractions to recover the analytes from their matrices, followed by solid phase separation to free the compounds from the background matrices of the substrates. Good limit of detection (LOD) and limit of quantification (LOQ) were achieved for the method used. The results of the validation study indicated that the method could be used to evaluate PPCP residues in the environment. The major contribution of this work is to further the understanding of the distribution and fate of PPCPs in the environment, which can assist in formulating relevant policy and management recommendations in South Africa. Future studies should focus on the level of PPCPs in underground water.

## Figures and Tables

**Figure 1 molecules-25-03114-f001:**
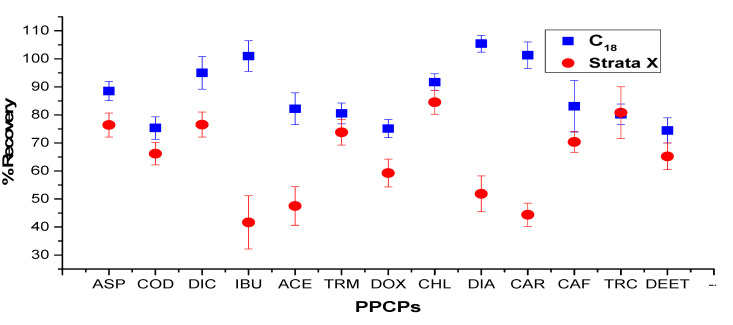
Recovery studied of the two selected cartridges in the 13 pharmaceuticals and personal care product PPCPs.

**Figure 2 molecules-25-03114-f002:**
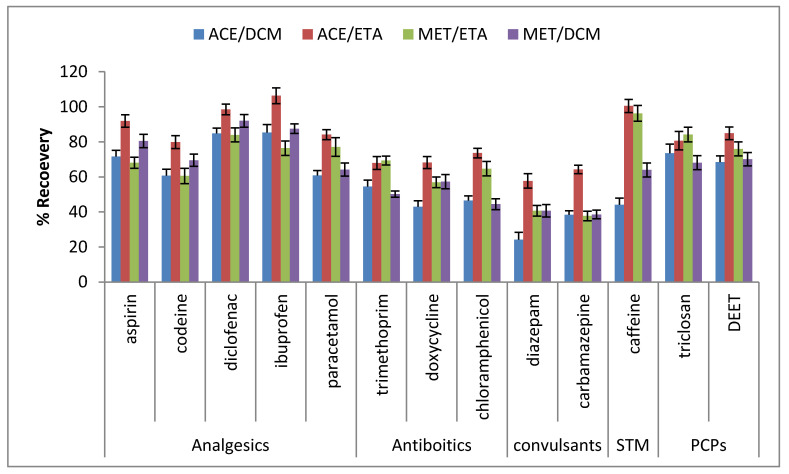
Solvents used for extraction in soil sample and % recovery.

**Figure 3 molecules-25-03114-f003:**
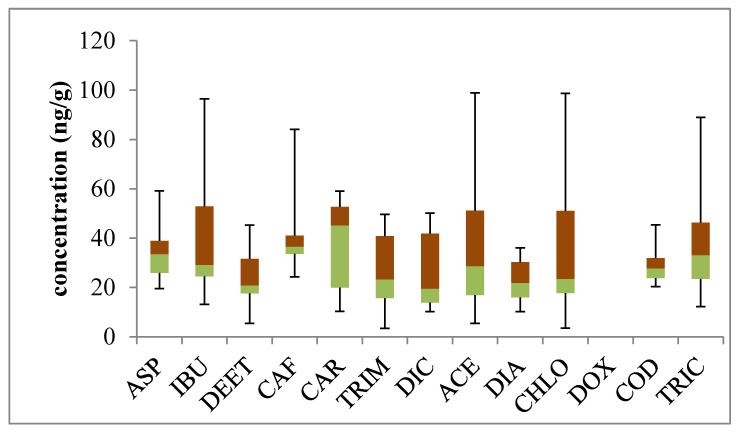
Box and whisker plot represents the mean concentration range (lowest, 25 percentile, median, 75 percentile and highest) of the PPCPs in the three sewage sludges.

**Figure 4 molecules-25-03114-f004:**
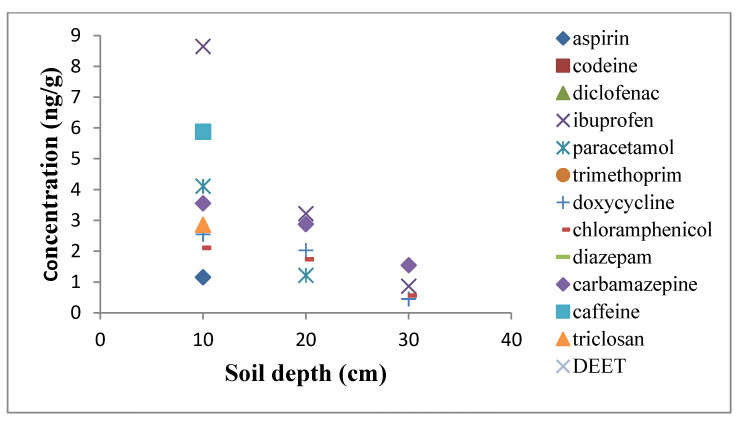
Concentration profiles of the examined compounds in soil (golf–course irrigated with effluents of the WWTP).

**Table 1 molecules-25-03114-t001:** Compounds, range of concentration, regression coefficients, limit of detection, and limit of determination of this finding and others the two substrates with their % recovery.

Compounds	Range (ng·g^−1^)	R^2^	Soils (ng·g^−1^)		Sludge (ng·g^−1^)		Others Findings in Soils (ng·g^−1^)		Others Finding in Sludge (ng·g^−1^)		Recovery (%)	
			LOD	LOQ	LOD	LOQ	LOD	LOQ	LOD	LOQ	sludge	soil
Aspirin	10–400	0.9949	0.1	0.3	0.2	0.5	0.14 *	0.38 *	1.1 *	3.6 *	89 ± 5 *	103 ± 4.2 *
Codeine	10–400	0.9994	0.3	0.8	0.1	0.3	na	na	na	na	98 ± 4 **	91 ± 13 **
Diclofenac	10–400	0.9919	0.5	1.6	0.8	2.6	0.16 *	0.48 *	0.7 *	2.3 *	98 ± 7 *	104.4 ± 3.3 *
Ibuprofen	10–400	0.9994	0.2	0.6	0.2	0.5	0.07 *	0.21 *	1.0 *	3.3 *	95 ± 4 *	104.4 ± 3.4 *
Paracetamol	10–400	0.9922	0.6	1.9	0.9	2.8	0.07 *	0.24 *	2.5 *	8.3 *	92 ± 13 *	86.2 ± 4.7 *
Chloramphenicol	10–400	0.9945	0.4	1.2	0.7	1.7	0.8 ****	2.7 ****	na	na	93 ± 5 ****	96 ± 5 ****
Doxycycline	10–400	0.9991	1.4	4.6	1.7	5.1	0.80 ***	2.67 ***	na	na	68 ± 8 **	62 ± 10 **
Trimethoprim	10–400	0.9922	0.2	0.6	0.1	0.4	0.64 ***	2.15 ***	na	na	97 ± 7 **	105 ± 5 **
Caffeine	10–400	0.9959	0.3	1.1	0.2	0.6	0.09 *	2.1 *	1.7 *	5.5	99 ± 11 *	98 ± 6 *
Carbamazepine	10–400	0.9985	0.3	1.0	0.6	1.7	0.16 *	0.44 *	1.5	5.0	98 ± 7 *	82 ± 10 *
Diazepam	10–400	0.9984	0.4	1.3	0.7	2.2	na	na	na	na	85 ± 3 **	79 ± 8 **
DEET	10–400	0.9975	0.5	1.5	1.2	3.3	0.89 *	1.48 *	0.58 *	1.31 *	86 ± 5 *	89 ± 6 *
Triclosan	10–400	0.9916	0.3	0.9	0.4	1.1	0.1 *	0.3 *	2.1 *	7.2 *	91 ± 4 *	91 ± 6.6 *

LOD = limit of detection. LOQ = limit of quantification. * [20], ** [21], *** [22], **** [16], na = not available.

**Table 2 molecules-25-03114-t002:** Mean recoveries (%) and standard deviation (*n* = 6) of the target compounds in the spiked soil and sludge samples.

Compounds	Leveling Spiking (Soils) Recovery (%)			Leveling Spiking (Sludge) Recovery (%)		
	50 ng	100 ng	200 ng	50 ng	100 ng	200 ng
Aspirin	88.5 ± 5.1	95.4 ± 8.1	81.6 ± 7.2	78.3 ± 4.5	74.4 ± 6.4	82.3 ± 5.8
Codeine	71.2 ± 4.3	68.4 ± 3.8	75.5 ± 5.3	64.7 ± 3.6	67.8 ± 3.9	71.3 ± 3.9
Diclofenac	101.5 ± 6.3	106 ± 2.8	106 ± 6.4	88.5 ± 5.7	84.1 ± 4.3	89.4 ± 5.6
Ibuprofen	101 ± 7.3	96.8 ± 7.6	89.2 ± 5.4	87.2 ± 2.0	89.4 ± 4.7	78.5 ± 6.4
Paracetamol	88.4 ± 4.5	82.3 ± 3.2	76.9 ± 2.2	67.5 ± 3.2	62 ± 2.1	69.5 ± 3.4
Chloramphenicol	54.5 ± 4.8	62.7 ± 5.1	67.4 ± 2.7	55.4 ± 2.5	63.5 ± 5.5	66.4 ± 8.6
Doxycycline	71.1 ± 3.1	77.3 ± 3.4	79.5 ± 7.8	67.5 ± 5.2	72.6 ± 7.8	65.4 ± 4.5
Trimethoprim	54.5 ± 4.3	63.6 ± 8.1	49.8 ± 7.2	51.5 ± 3.3	55 ± 3.2	49.6 ± 2.1
Caffeine	89.6 ± 3.6	94.6 ± 1.9	102.7 ± 2.9	76.5 ± 2.5	77.9 ± 4.1	69.2 ± 1.8
Carbamazepine	56.6 ± 5.0	53.1 ± 3.4	60.7 ± 4.9	61.5 ± 4.9	63.2 ± 2.1	59.5 ± 1.1
Diazepam	45.6 ± 4.2	48.5 ± 3.8	53 ± 5.0	57.5 ± 4.4	60.1 ± 5.1	61.4 ± 4.2
DEET	60.4 ± 4.1	65.3 ± 3.3	70.2 ± 2.3	56.8 ± 4.1	53.1 ± 2.9	64.3 ± 7.6
Triclosan	76.2 ± 3.1	83.5 ± 6.9	72.6 ± 3.0	70.5 ± 4.5	74.5 ± 5.5	67.8 ± 3.2

**Table 3 molecules-25-03114-t003:** Precision data of the extraction procedures for the two types of substrates.

Compounds	Soil		Sludge	
	Repeatability RSD (%) *n* = 6	Intermediate Precision RSD (%) *n* = 3, *k* = 3	Repeatability RSD (%) *n* = 6	Intermediate Precision RSD (%) *n* = 3, *k* = 3
Aspirin	3.0	4.5	3.5	4.5
Codeine	8.5	9.0	11	13
Diclofenac	4.0	5.0	4.0	5.5
Ibuprofen	3.5	5.5	3.3	5.8
Paracetamol	2.0	3.0	2.0	3.5
Chloramphenicol	10	11.5	12	13
D oxycycline	8.5	9.5	9.2	9.5
Trimethoprim	7.0	8.5	8.5	9.0
Caffeine	5.0	5.5	5.5	8.0
Carbamazepine	5.0	6.0	4.5	6.0
Diazepam	4.5	5.0	5.0	7.5
DEET	3.5	4.5	4.4	4.5
Triclosan	3.0	4.5	4.0	5.7

**Table 4 molecules-25-03114-t004:** Therapeutic groups, compounds/IUPAC names and structures of 13 PPCPs studied.

	Therapeutic Groups/Abbreviation	Compounds/IUPAC Names	Structures
1	Analgesic/anti-inflammatory/ASP	Aspirin/2-Acetoxybenzoic acid	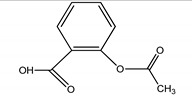
	COD	Codeine/ (5a, 6a) -3-Methoxy-17-methyl-7	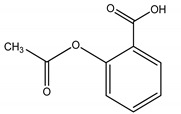
	DIC	Diclofenac/2-[2-(2,6-dichloroaniline) phenyl] acetic acid	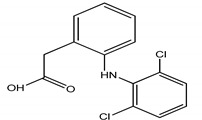
	IBU	Ibuprofen/2, (4-isobutylphenyl) propanoic acid	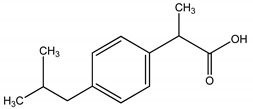
	ACE	Paracetamol/*N*-(4-hydroxphenyl) ethanamide	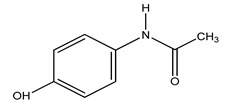
2	Antibiotics/CHL	Chloramphenicol/2,2-dichloro-*N*-(1,3-dihydroxy-1-) 4-nitrophe	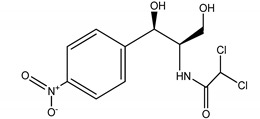
	DOX	Doxycycline/Pentahyroxy-6-methyl-1,11-dioxo	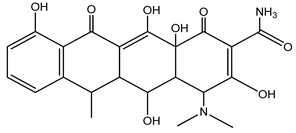
	TRM	Trimethoprim/5- (3,4,5-trimethoxybenzyl)-2, 4-pyrimidinediamine	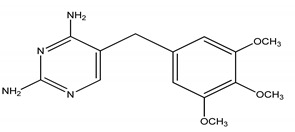
3	Anticonvulsants/CAR	Carbamazepine/5H-Dibenzo [b,f] azepine-5-carboxamide	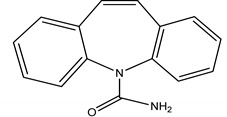
	DIA	Diazepam/7-chloro-1-methyl-5-phenyl-1,3-dihydro-2*H*-1,4-benzediazepin 2-one	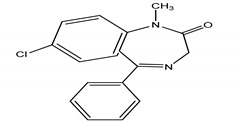
4	Stimulant/CAF	Caffeine/1,3,7-trimethylpurine-2,6-dione	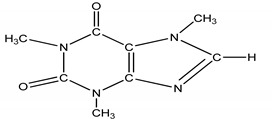
5	PCPs/TRC	Triclosan/5-chloro-2- (2,4-dichlorophenoxyl) phenol	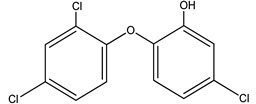
	DEET	DEET/*N*,*N*-Diethyl-meta-toluamide	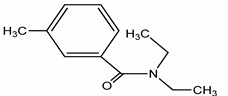

**Table 5 molecules-25-03114-t005:** Results and properties of acetylation and silylation derivatization of the 13 analytes; retention time (RSD%, *n* = 6), CAS numbers, ions, and pka, log Kd.

Acetylation					Silylation			
Compound Names	RT (RSD%)	CAS Number	MW: m/z Ions	Pka: logkd	Compound Names	RT (RSD%)	CAS Number	MW: m/z Ions
**Analgesic/Anti-Inflammatories**								
Aspirin	10.470 (0.9)	00050-78-2	180:120, 180, 43	3.5:1.19	Aspirin	10.412 (0.2)	00050-78-2	252:120, 115, 210
2H-indol-2-one	19.382 (0.77)	015362-40-0	277:214, 242, 277	4.14:4.51	Diclofenac	21.256 (0.3)	959106-20-8	367:214, 242
Ibuprofen	11.77 (0.8)	061566-34	206:109, 161, 206	4.91:3.97	Ibuprofen	11.292 (0.3)	015687-27-1	278:109, 161, 234
Morpian-6-ol	24.663 (0.6)	006703-27-1	341:341, 282, 229	5.0:0.48	Codeine	23.840 (0.5)	074367-14-9	299:299, 162, 229
Acetaminophen	13.737 (0.9)	000103-90-2	194:109, 151, 194	9.38:0.46	Paracetamol	13.423 (0.4)	041571-82-8	295:116, 206, 280
**Antibiotics**								
Chloramphenicol	n.d.			9.61:n/a	Chloramphenicol di (trimethylsily)	24.988 (0.4)	1000386-63-9	466, 225, 208, 242
Trimethoprim	n.d.			1.5:0.59	Trimethoprim	25.560 (0.3)	000738	290:290, 259, 275
Doxycycline	n.d.				Doxycycline	15.992 (0.4)		270:167, 255, 58.1
**Anticonvulsants**								
Diazepam	23.734 (0.9)	00439-14-5	284:256, 283, 221	0.10:2.8	Diazepam	23.733 (0.4)	00439-14-5	284:256, 283, 221
5-Acety-5H-dibenz (b,f) azepine	18.99 (0.9)	015362-40-0	235:193, 235, 165	7:2.47	Carbamazepine	20.993 (0.3)	000298-46-4	308:193, 235, 293
**Stimulant**								
Caffeine	14.612 (0.9)	00058-08-2	194:194, 109, 67	2.0:−0.63	Caffeine	14.809 (0.3)	00058-08-2	194:194, 109, 67
**PCPs**								
Phenol, 5-chloro-2-	19.522 (0.98)	004623	330:288, 218, 146	−0.95:4.8	Triclosan	19.499 (0.3)	003380-34-5	362:347, 218, 310
DEET	10.663 (0.9)	000134-62-3	199:190, 119, 91		DEET	10.659 (0.4)	000134-62-3	199, 190, 119, 91

n.d. = not detected, n/a = not available RT = Retention time.

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
