# Peer review of "Method Validation and Investigation of the Levels of Pharmaceuticals and Personal Care Products in Sludge of Wastewater Treatment Plants and Soils of Irrigated Golf Course"

_molecules, 2020, doi:10.3390/molecules25143114_

Round 1

Reviewer 1 Report

The manuscript proposed here by Olufemi T. Ademoyegun et al. describes the development, validation and application of a new GC-MS method for the analysis of pharmaceutical products and personal care (PPCP) in sludge and soil samples for environmental purposes. The methodology was developed and validated to simultaneously detect 13 different analytes (5 analgesics, 3 antibiotics, 2 anticonvulsants, 1 stimulant and 2 personal care products) in the two matrices and is applied for the analysis of real samples of sludge of waste water treatment plant and soils of irrigated golf courses. The strategies and data reported in this manuscript could be of interest to the environmental science and analytical scientists.

However, there are some issues that need to be addressed before the manuscript can be considered for publication.

In the "Introduction" section, line 67, there is an incorrect reference.

The grammar and English language must be extensively checked throughout the manuscript and tables. (e.g. anticonvulsants in Table 1, singular/plural terms in Table 2, etc.). 

Table 1 is difficult to understand, the layout and the information reported are organised in an unclear way and should be revised in order to make it understandable.

The chemical structures of the analytes are not reported in the text. They should be reported in the "Introduction" or in the "Experimental" sections.

In sub-paragraph 2.1, the IUPAC names of the analytes are missing and it is not clear where methanol, dichloromethane, ethyl acetate, acetone and h-hexane are purchased. These sections need to be implemented.

In sub-paragraph 2.3, on line 128, there is something wrong with the sentence, which must be corrected. Another doubt concerns the pH of Milli-Q grade water, has it been measured? Because nothing is reported in the text on instruments used for pH measurement. It is not clear how the Authors can state the pH is 7 if it has not been measured or corrected.

In sub-paragraph 2.5 it is not clear how the transitions and MS parameters for each analyte were chosen for the analysis. This section needs to be implemented.

Section 2.6 shows the results obtained regarding the calibration curves, LOD and LOQ for both matrices. But these are results, they shouldn't be reported in the "Experimental" section. The Authors should implement sub-paragraph for these data under the "Results and discussion" paragraph.

Regarding the paragraph "Results and discussion", there are several bibliographical references reported together with the results obtained. It may be useful for the discussion to report only some of these. This Reviewer believes some comments should be summarized or moved before this paragraph (for example, the part of text from line 244 to line 253, in paragraph 3.1).

The manuscript also lacks the references for analytical guidelines used for method validation and some representative chromatograms of the analysis carried out, in particular, for the analysis of real samples (including blank and LOQ chromatograms) should be shown.

The Reviewer believes that the manuscript needs a minor revision in order to implement the information and data obtained in the study and to correct some errors and inaccuracies. After this revision work, the manuscript could be considered for publication in Molecules.

Author Response

Comment

Change

In correct reference

Change to (Santana et al., 2009; Azzouz et al., 2012; Bu et al., 2013).

Anticonvulsant in table now 2 and 3

Change to anticonvulsants

The structure and IUPAC name of the analytes

Added, form table 1.

All solvent used

purchased from (Merck, Darnstadt, Germany)

pH of milligrade

Removed

Line 244 to 252 now

In this study, the recovery efficiency of the target analytes were assessed with the aforementioned combined solvents, and we observed that solvents like acetone-methanol, dichloromethane-methanol and methanol-water combinations could also recover most of the target analytes satisfactorily. However, their recovery percentages were lower than with acetone-ethyl acetate combination (Figure 2). Some compounds like trimethoprim and doxycycline recorded poor recoveries as shown in Figure 2, combination of acetone and ethyl acetate which yielded the optimum recovery for the selected compounds was therefore used for other extractions in this study. The average recoveries of the analytes extracted with this solvents mixture ranged from 57.7 – 106.3% at the 200 ng spiked concentration, which is a bit concur to the results gotten by Xu et al., (2008) which ranged from 63.8 to 110.7% and slightly better recovery than Gatidou et al., (2007) result performance results that ranged from 47.6 – 106%. Others research studies for optimization of solvents, probing the recovery rates for extracting target compounds from soil samples using different solvents. Their studies for natural soils and sediments spiked with PPCP mixture using microwave assisted solvent extraction with solvent mixture, methanol: water (3:2) vol/vol indicate d optimum recovery rate. (Azzouz et al., 2012; Bu et al., 2013). In addition, Xu et al., (2008) indicated in their study that the mixture of acetone and ethyl acetate produced better recovery for the PPCPs and some endocrine disruptive chemicals (EDCs) in sediments. Combination of acetone with methanol (1: 1) vol/vol, also yielded percentage recoveries in the range of 47.6 – 106%, and which were best superlative recoveries for the selected compounds (Gatidou et al., 2007). Agunbiade and Moodley (2016) also reported combined acetone with 10% acetic acid - ethyl acetate solvents as best for the extraction of target compounds in the solid sample.

LOD and LOQ

Now included in the results and discussion.

Reviewer 2 Report

The subject is interesting and methodology is in accordance with modern trends in analytical chemistry. The method is well validated and the determination of analytes in real samples are done. I have some editorial remarks.

Check the way of writing dimension for degrees Celsius.

v. 152 space between % and RSA

Table 3 - spaces between number and dimension

Literature The lack of pages in Alvarino et al., Azzous et al. (v. 404) and Huerta and al.

Author Response

Reviewer 2.

Comment

                                                Change

Writing of degree celsius

Change in all the manucripts e.g 180 0C

Space between % and RSA

EFFECTED IN LINE 155

Lack page number

Inserted in line 388, pp 701-709, line 402, pp 231-241, line 440, pp 241-249.

Reviewer 3 Report

The manuscript entitled: 'Method validation and investigation of the levels of pharmaceuticals and personal care products in sludge of wastewatter treatment plants and soils of irrigated golf course' presents a very interestic subject. It is well written and documented. However English needs improvement especially in the experimntal part. 

In the abstract it is written that satisfactory recoveries > 51,5% have been achieved and in the Table 3 some compounds have recoveries of 45.6 % and 49.6%. Please clarify. 

The authors should also provide a Table with the recoveries achieved by their experimantal work and the existing in the literature for comparison.

Additionally the LOQ's/compound achieved by the authors must be compared with the one's in the literature.

The advantage of the proposed method with other published methods must be written clearly in the conclusion 

Also units e.g ng/g must be written as ng g-1

Please do all the correction in the text.

Author Response

Reviewer 3.

Comment

                                          Change

  From 51.5 %

satisfactory recoveries (> 47.6%) have change minimum recovery in the table

The conclusion have change to

The validated method gave acceptable recovery efficiencies for all the target analytes in the two different substrates matrixes. It provided an analytical tool to study the presence of selected compounds in any solid matrix in the environment. The presence of PPCPs in the two substrates were simultaneously determined by ultrasonic assisted solvent extractions to recover the analytes from its matrices, followed by solid phase separation to free the compounds from the background matrices of the substrates. Good limit of detection (LOD) and limit of quantification (LOQ) were achieved for the method used. The results of the validation study indicated that the method could be used to evaluate PPCPs residues in environment. The major contribution of this work is to further the understanding of the distribution and fate of PPCPs in the environment, which can assist in formulating relevant policy and management recommendations in South Africa. Future studies should focus on the level of PPCPs in underground water.

ng/g

 All change to ngg-1     and others similar issues

Round 2

Reviewer 3 Report

No further comments

Author Response

others research findings on LoD, LOQ, and % recovery have been added to the table 3. Thanks.
